# Hybrid Nanocomposite Platform, Based on Carbon Nanotubes and Poly(Methylene Blue) Redox Polymer Synthesized in Ethaline Deep Eutectic Solvent for Electrochemical Determination of 5-Aminosalicylic Acid

**DOI:** 10.3390/s21041161

**Published:** 2021-02-07

**Authors:** Oana Hosu, Madalina M. Barsan, Robert Săndulescu, Cecilia Cristea, Christopher M. A. Brett

**Affiliations:** 1Department of Chemistry, CEMMPRE, Faculty of Sciences and Technology, University of Coimbra, 3004-535 Coimbra, Portugal; hosuoanaalexandra@gmail.com (O.H.); madalina.barsan@gmail.com (M.M.B.); cbrett@ci.uc.pt (C.M.A.B.); 2Department of Analytical Chemistry, Faculty of Pharmacy, “Iuliu Haţieganu” University of Medicine and Pharmacy, 400349 Cluj-Napoca, Romania; rsandulescu@umfcluj.ro; 3National Institute of Material Physics, 077125 Magurele, Romania

**Keywords:** ethaline deep eutectic solvent, 5-aminosalicylic acid, poly(methylene blue), nanostructured polymer film, electrochemical sensor

## Abstract

A novel hybrid composite of conductive poly(methylene blue) (PMB) and carbon nanotubes (CNT) was prepared for the detection of 5-aminosalicylic acid (5-ASA). Electrosynthesis of PMB with glassy carbon electrode (GCE) or with carbon nanotube modified GCE was done in ethaline deep eutectic solvent of choline chloride mixed with ethylene glycol and a 10% *v/v* aqueous solution. Different sensor architectures were evaluated in a broad range of pH values in a Britton-Robinson (BR) buffer using electrochemical techniques, chronoamperometry (CA), and differential pulse voltammetry (DPV), to determine the optimum sensor configuration for 5-ASA sensing. Under optimal conditions, the best analytical performance was obtained with CNT/PMB_DES_/GCE in 0.04 M BR buffer pH 7.0 in the range 5–100 µM 5-ASA using the DPV method, with an excellent sensitivity of 9.84 μA cm^−2^ μM^−1^ (4.9 % RSD, *n* = 5) and a detection limit (LOD) (3σ/slope) of 7.7 nM, outclassing most similar sensors found in the literature. The sensitivity of the same sensor obtained in CA (1.33 μA cm^−2^ μM^−1^) under optimal conditions (pH 7.0, *E*_app_ = +0.40 V) was lower than that obtained by DPV. Simultaneous detection of 5-ASA and its analogue, acetaminophen (APAP), was successfully realized, showing a catalytic effect towards the electro-oxidation of both analytes, lowering their oxidation overpotential, and enhancing the oxidation peak currents and peak-to-peak separation as compared with the unmodified electrode. The proposed method is simple, sensitive, easy to apply, and economical for routine analysis.

## 1. Introduction

Deep eutectic solvents (DES) are generally composed of at least one hydrogen bond donor (HBD) and a hydrogen bond acceptor (HBA). This mixture particularly resembles a self-melting property as, when put in contact, they form an eutectic phase with a melting point lower than that of each component individually [1,2]. DES have been shown to overcome the most important pitfalls of ionic liquids, i.e., predecessors of DES, such as high toxicity, cost, and non-biodegradability [3]. As a result, scientists have gained more interest in using eutectic solvents in various research domains due to their easily tailored and non-toxic “green” solvent properties [4,5]. Wide-ranging operational windows for electrochemical procedures coupled with DES intrinsic ionic features, ease of alloy, and semiconductor electrodeposition are now available. The combined use of cosolvents, counterions, or applied potential are, as yet, under-explored strategies for tailoring electrode surface interactions leading to controlled film synthesis [6].

Recently, multiple ethylene glycol (EG)-based eutectic solvents were synthesized for electrochemical applications [7]. The physical properties of these DES such as density, viscosity, conductivity, and surface tension are greatly influenced by the ratio of the EG (HBD) and the nature and concentration of the HBA [8]. All these properties along with coordination with solutes or other surfaces play noteworthy roles in modulating the synthesis of nanostructures.

Conducting polymers, especially electroactive polymers, have been used, in recent years, to develop new polymeric materials, with significantly good electrical and optical properties and high conductivity/weight ratio that exhibit biodegradable, biocompatible, and porous properties [9,10,11]. In addition, the synthesis of polymers in DES media can lead to the formation of films in which the DES components are able to modulate nucleation and growth mechanisms by charge neutralization and shifts in oxidation/reduction potentials, promoting growth in optimal nanoscale-sized systems [4]. It has been shown that the addition of counterions in aqueous solutions, at a low percentage, highly influences the conductivity of polymers synthesized in DES as compared with their analogues, either in pure DES or in aqueous media [12], tailoring the monomer reactivity and mass transport; in this way, the electrochemical stability and reactivity of EG can be increased. For polymer electrosynthesis, parameters such as surface tension, polarity, and viscosity have a strong influence on the structure of the films [13]. This has been shown for poly(3,4-ethylenedioxythiophene) [14,15], polyaniline [16,17,18], 3-aminophenylboronic acid [19], polypyrrole [20,21], poly(acrylic acid) [22], poly(ethylene terephthalate) [23], poly(brilliant cresyl blue) [24], poly(thionine) [25], poly(methylene green) [26], and poly(neutral red) [27], mainly because of their control of monomer reactivity and mass transport properties.

Another strategy to obtain highly performing electroconductive platforms is the integration of nanomaterials to form polymer composites. Nanomaterials such as carbon nanotubes, graphene, metallic nanoparticles, or magnetic particles have been introduced in the synthesis of polymer-based materials to obtain hybrid composites with synergetic properties [28]. The use of deep eutectic solvents for the development of new nanomaterials has shown great success as an alternative in the synthesis of conjugated conducting polymers [4,29], DNA/RNA architectures, metal-organic frameworks, electrodeposited films, colloidal assemblies, hierarchically porous carbons [6,30]. Recently, the electropolymerization procedure for poly(methylene blue) (PMB) synthesis in DES was optimized [12] and combined with carbon nanotubes (CNT) to develop new platforms for electrochemical sensors [31].

5-Aminosalicylic acid (5-ASA, mesalazine, or mesalamine) is one of the non-steroidal anti-inflammatory drugs, structurally related to both acetaminophen and acetyl salicylic acid. It is widely prescribed for the treatment of inflammatory bowel disease (IBD) such as ulcerative colitis and Crohn’s disease and prevention of inflammation-associated colorectal cancer (CRC) [32,33]. 5-ASA is a prodrug of sulfasalazine, both being used for the same purpose. Although, sulfasalazine has superior therapeutic activity against ulcerative colitis, in a later publication, it was highlighted that oral 5-ASA formulations present inferior side effects as compared with sulfasalazine [34]. However, the overdose or long-term use of 5-ASA can severely affect the physiological functions of body systems such as renal, hepatic, pancreas, respiratory, cardiac, and immune system [35]. Therefore, it is of high importance to develop simple and cost-effective tools for the monitorization of 5-ASA in pharmaceutical formulations or biological samples to ensure high performance quality control and clinical analysis procedures.

In this work, an electrochemical method was developed and successfully applied for the detection of 5-ASA by using an electrochemical platform composed of a nanostructured phenazine polymer electrochemically synthesized in ethaline (PMB_DES_) combined with multiwalled carbon nanotubes (CNT). First, the oxidation mechanism of 5-ASA was determined, and the experimental conditions optimized for electrochemical detection by chronoamperometry (CA) and differential pulse voltammetry (DPV). Various electrode architectures based on PMB_DES_ and CNT were evaluated with only one, or two components, in order to evaluate the contribution of each component and to find the optimal electrode configuration for 5-ASA determination in pharmaceutical samples and simultaneously with its analogue, acetaminophen.

## 2. Materials and Methods

### 2.1. Reagents and Solutions

All reagents were of analytical grade and were used without further purification. Millipore Milli-Q nanopure water (resistivity ≥ 18 MΩ cm) was used for the preparation of all solutions.

5-Sminosalicylic acid (5-ASA), acetaminophen (APAP), methylene blue (MB), choline chloride, ethylene glycol, potassium chloride, sodium monobasic phosphate, sodium dibasic phosphate, sodium hydroxide, acetic, boric, hydrochloric, perchloric, and phosphoric acid were purchased from Sigma-Aldrich (Darmstadt, Germany).

For the electrochemical characterization of the modified electrodes, the supporting electrolyte was 0.04 M Britton Robinson (BR) buffer solution, pH 7.0. All experiments were carried out at room temperature (25 ± 1 °C).

### 2.2. Instrumentation

Electrochemical experiments were performed in a three-electrode cell (2 mL), containing the glassy carbon electrode (GCE, geometric area 0.00785 cm^2^) as the working electrode, a Pt wire counter electrode, and an Ag/AgCl (3.0 M KCl) reference electrode (Metrohm Autolab, Utrecht, The Netherlands), using a potentiostat/galvanostat µ-Autolab system (Metrohm Autolab, Utrecht, The Netherlands). Before each use, the surface of the GCE was cleaned with diamond spray and polishing paper (Kemet, High Wycombe, UK).

The pH measurements were carried out with a CRISON 2001 micro pH-meter (Crison Instruments SA, Barcelona, Spain) at room temperature.

### 2.3. PMB_DES_ and CNT-Based Modified GCE

The hybrid composite, based on multiwalled CNT and PMB in ethaline (PMB_DES_) modified GCE, was prepared, in accordance with the optimized procedure reported in our previous work [31]. Briefly, two layers of 1 µL of 1% HNO_3_-CNT dispersion in chitosan solution (1.0% m/m chitosan dissolved in 1.0% *v*/*v* acetic acid) [36] were deposited by drop coating on the surface of GCE and allowed to dry at room temperature. Thereafter, the CNT/GCE was electrochemically modified by potential cycling (−0.6–1.1 V, 150 mV s^−1^ for 30 scans) polymerization in a solution containing 5 mM MB in DES (ethaline = choline chloride (1): ethylene glycol (2)) with an additional 10% aqueous mixture of 0.1 M NaOH and HClO_4_ to lead to the formation of PMB_DES_/CNT/GCE. To form CNT/PMB_DES_/GCE, polymerization was done before drop coating with CNT [12].

The different electrode architectures studied for 5-ASA detection were the following: bare GCE, PMB_DES_/GCE, CNT/GCE, PMB_DES_/CNT/GCE, and CNT/PMB_DES_/GCE.

### 2.4. Electrochemical Measurements

Electrochemical characterization by cyclic voltammetry (CV), chronoamperometry (CA), and differential pulse voltammetry (DPV) was conducted. Cyclic voltammograms (CVs) were recorded in 0.1 M KCl + 5 mM HCl in the potential range from −0.4 to 1.0 V at 50 mV s^−1^. The CA measurements were done at a fixed potential of + 0.40 V vs. Ag/AgCl at different platforms based on PMB_DES_ and CNT for 5-ASA determination in 0.04 M BR buffer at pH 7.0. The DP voltammograms were recorded by scanning the potential from 0.0 to +0.50 V with a scan rate of 4 mV s^−1^, 2 mV step potential, and 50 mV pulse amplitude. To determine the oxidation mechanism of 5-ASA and analyzing sensor performance by both CA and voltammetric methods, a pH study was conducted involving 5-ASA detection in 0.04 M BR buffer in the pH range from 2.0 to 8.0.

### 2.5. Analytical Applications

Standard solutions of 5-ASA and APAP were prepared and kept in the refrigerator for further use. All 5-ASA and APAP solutions for testing were prepared by diluting the stock solutions to the desired concentration. To ensure the dissolution and improved stability of 5-ASA, the stock solution was prepared in 0.5 M HCl, the pH of the diluted solutions being thereafter adjusted in BR buffer.

Twenty PENTASA^®^ tablets (Ferring GmbH containing 500 mg 5-ASA/tablet) were finely ground and an accurately weighed quantity of powder was dissolved in 50 mL distilled water with the addition of 5 mL of 5 M HCl and sonicated for 20 min. After centrifugation, the removed supernatant was diluted to the desired concentration and sample aliquots of approximately twice the concentration of the standard were added, applying the standard addition method.

Coldrex Max Grip^®^ (1 g acetaminophen, 10 mg phenylephrine hydrochloride, and 40 mg ascorbic acid) pharmaceutical sachets were dissolved in 50 mL distilled water. The analysis of 5-ASA and APAP was performed in a 2 mL cell, the CNT/PMB_DES_/GCE acting as a working electrode, by both CA and DPV methods, as previously described. Aliquots of approximately twice the concentration of that of the 5-ASA and APAP pharmaceutical formulations were added, and measurements made in triplicate, using the standard addition method, with four additions of standard solution.

## 3. Results and Discussion

### 3.1. Electrochemical Characterization of 5-Aminosalicylic Acid (5-ASA) by Cyclic Voltammetry and Differential Pulse Voltammetry

Figure 1 shows cyclic voltammograms (CVs) in 0.1 M KCl + 5 mM HCl with PMB_DES_/GCE in the absence and presence of 100.0 µM 5-ASA. It can be observed that PMB oxidation and reduction peaks both decrease upon 5-ASA addition, which might be due to the high concentration of 5-ASA present in the solution that can impede the redox reaction at the polymer interface. Moreover, the –COOH group in the 5-ASA structure is prone to form hydrogen bonds with the nitrogen atoms in the polymer film [37]. This interaction has two results as follows: (i) increases the 5-ASA concentration at the PMB surface, which improves the 5-ASA reaction kinetics and implicitly the sensitivity of the method and (ii) hinders the electrochemical process of PMB, due to a decrease in counter ion density/availability at the polymer/solution interface. Furthermore, a new oxidation peak at *E* = +0.64 V appears, corresponding to 5-ASA oxidation (16.0 µA cm^−2^) demonstrating the possibility of electrochemically detecting 5-ASA with the PMB_DES_ based sensor.

The pH of the supporting electrolyte plays an important impact on the anodic peak currents and peak potentials; therefore, Figure 2 presents the electrochemical oxidation of 50.0 μM 5-ASA with GCE and PMB_DES_/GCE assessed by DPV in BR buffer solutions in the pH range from 2.0 up to 8.0 in order to determine the oxidation mechanism (Figure 2a) and to obtain the best peak resolution and maximum sensitivity towards 5-ASA. With an increase in pH, Figure 2c,d, the oxidation peak potential, *E*_pa_, shifts linearly in the negative direction according to the following equation: *E*_pa_/mV = 624 − 56.5 pH (*R**^2^* = 0.9830) and *E*_pa_/mV = 622 − 61.2 pH (*R**^2^* = 0.9986) with GCE, and at PMB_DES_/GCE, respectively. As both slopes are very close to the theoretical value of –58 (*m*/*n*) mV per pH unit, where *m* represents the number of protons and *n* represents the number of electrons, it can be determined that the number of H^+^ involved in the oxidation process of 5-ASA is equal to the number of e^−^ [38]. Concerning the anodic peak potential of 5-ASA, there is a shift towards more negative values from +0.51 V at pH 2.0 to +0.15 V at pH 8.0. In acid media, the amino group of 5-ASA is protonated and hinders the oxidation process, which requires higher potential values. Above pH 6, the oxidation of 5-ASA occurs at a lower potential, as the -NH_2_ group is no longer protonated (p*K*_a2_ = 5.8) [39]. No influence of the 5-ASA electrochemical process was observed when the PMB oxidation mechanism was investigated (Appendix A).

The value of the DPV peak width at half-height *W*_1/2_, is 61 mV and 57 mV at CNT/PMB_DES_/GCE and PMB_DES_/CNT/GCE, respectively. The theoretical minimum value is 45 mV for a reversible electrochemical reaction involving the transfer of two electrons [38]. However, the theoretical width of the DPV response for a slow transfer can increase to ca. two-fold [40], as also seen for GCE (87 mV) and PMB_DES_/GCE (125 mV). Therefore, considering the Δ*E*/ΔpH slope values and *W*_1/2_, it was concluded that 5-ASA oxidation occurs with a two electron and two proton transfer, as also found in other 5-ASA reported papers [41,42]. A possible oxidation mechanism is displayed in Scheme 1.

The highest peak current value for 5-ASA oxidation was obtained in pH 2.0 solution at +0.52 V, while at pH 7.0 (*E*_pa_ = +0.18 V) the peak current is 50% of that in pH 2.0 buffer, but occurs at a much lower potential, being more suitable for pharmaceutical analysis since fewer electroactive molecules in complex samples interfere at this potential. In addition, the pH of blood and many other body fluids is close to 7.0, and therefore pH 7.0 (0.04 M BR buffer) was selected as the best supporting electrolyte for the electrochemical determination of 5-ASA in pharmaceutical formulations.

A comparison of the 5-ASA response in BR buffer at pH 7.0 at different electrode architectures is displayed in Figure 2b. With a bare GCE, a small oxidation peak current of 34.5 μA cm^−2^ was observed, slightly increasing with PMB_DES_/GCE, being shifted by 70 mV towards more negative potential values, indicating a catalytic effect from the PMB polymer towards 5-ASA oxidation. The presence of CNT in the electrode architecture led to a significant increase in the oxidation peak values by a factor of 5.5 (*j*_pa_ = 194 μA cm^−2^) and 11.6 (*j*_pa_ = 405 μA cm^−2^) for PMB_DES_/CNT/GCE and CNT/PMB_DES_/GCE, respectively, which can be explained by the increased electroactive area of the modified electrode. 

### 3.2. Electrochemical Detection of 5-ASA

#### 3.2.1. Chronoamperometry

##### Optimization of Experimental Conditions: pH and Applied Potential

Considering the oxidation peak potential of 5-ASA with PMB_DES_/GCE, the CA measurements were first performed at *E* = +0.60 V. The effect of pH on the CA response of 50 µM 5-ASA with PMB_DES_/GCE was evaluated at different pH values in the range from 2.0 to 8.0 using 0.04 M BR buffer solutions as supporting electrolyte (Figure 3a). Due to the presence of the carboxylic group (-COOH, p*K*_a1_ = 3.0), primary aromatic amino group (-NH_2_, p*K*_a2_ = 5.8), and phenolic group (-OH, p*K*_a3_ = 13.9) in the molecule [43], the charge of 5-ASA is highly dependent on pH. As seen in Figure 3a, the highest currents were obtained at pH 2.0 and 7.0. The current at pH 3.0 is less by approximately 25% than that at pH 2.0, but thereafter, follows a positive trend up to the highest current at pH 7.0 which is 11% higher than that at pH 2.0. On the basis of these results, pH 7.0 was used in the subsequent experiments.

Secondly, the applied potential for 5-ASA determination was optimized. The CA response to 50 µM 5-ASA with PMB_DES_/GCE in BR buffer at pH 7.0 was analyzed at applied potential values ranging from *E* = +0.20 to +0.60 V. As seen in Figure 3b, the highest current density is obtained at *E* = +0.60 V and slightly decreases down to *E* = +0.30 V, and almost disappears at *E* = +0.20 V. Although the current density is higher at *E* = +0.60 V, in real sample analyses, the possible interference reactions at this potential are much higher than at lower applied potentials. Therefore, *E* = +0.40 V was chosen as the applied potential to ensure the anodic conversion of 5-ASA to its quinonoid form, whilst minimizing the matrix effect.

An overview of the CA response of 5-ASA at different pH values and applied potentials can be seen in Appendix A. The influence of pH and applied potential was also evaluated at sensors based on CNT together with PMB_DES_ (Appendix A).

##### Analytical Parameters of the Optimized Sensors

Different sensor architectures based on the phenazine polymer and CNT were evaluated for 5-ASA determination by CA at *E* = +0.40 V vs. Ag/AgCl using BR buffer, pH 7.0, as supporting electrolyte. The analytical parameters of the sensors are presented in Appendix A.

As can be seen in Appendix A, an increase in the sensitivity towards 5-ASA from bare GCE to one component-modified (PMB_DES_ or CNT) electrode to hybrid composite platforms (PMB_DES_-CNT) in the sequence GCE < PMB_DES_/GCE < CNT/GCE < PMB_DES_/CNT/GCE < CNT/PMB_DES_/GCE is observed, with CNT greatly improving sensor performance. Figure 4 shows the CA response in the concentration range 0.5–80 µM 5-ASA for CNT/PMB_DES_/GCE, and 10.0–80 µM 5-ASA for CNT/GCE and PMB_DES_/GCE, respectively. The best synergistic effect from PMB_DES_ together with CNT was observed when CNT were deposited on top of the polymer film (Figure 4b) with the corresponding regression equation Δ*j* (μA cm^−2^) = 1.33 [5-ASA] (μM) – 0.13 (R^2^ = 0.9998). The detection limit (LOD) was calculated as three times the standard deviation of the calibration plot divided by the slope of the calibration plot [44], obtaining a low LOD of 56.9 nM 5-ASA and a sensitivity of 1.33 μA cm^−2^ μM^−1^, by CNT/PMB_DES_/GCE sensor.

#### 3.2.2. Differential Pulse Voltammetry

The quantification of 5-ASA by DPV was carried out in a BR buffer solution, pH 7.0, as supporting electrolyte with PMB_DES_/CNT/GCE and CNT/PMB_DES_/GCE sensors, which exhibited the best analytical performance using the fixed potential technique. A sharp and well-defined oxidation peak appears at the peak potential of +0.17 and + 0.19 V in the presence of 5-ASA with PMB_DES_/CNT/GCE and CNT/PMB_DES_/GCE, respectively. Figure 5 illustrates DPVs with PMB_DES_/CNT/GCE and CNT/PMB_DES_/GCE in BR buffer (pH 7.0). A linear range of 5–100 µM was obtained for 5-ASA with sensitivities of 3.85 μA cm^−2^ μM^−1^ (RSD = 2.2%, *n* = 5) with PMB_DES_/CNT/GCE and 9.84 μA cm^−2^ μM^−1^ (RSD = 4.9%, *n* = 5) with CNT/PMB_DES_/GCE. The resulting detection limits for 5-ASA were 58.2 and 7.7 nM with PMB_DES_/CNT/GCE and CNT/PMB_DES_/GCE, respectively.

The analytical parameters of the developed 5-ASA electrochemical sensors were compared with other reported sensors, Table 1. The CNT/PMB_DES_/GCE sensor showed an overall superior performance to those found in the literature, presenting a significantly lower detection limit with high sensitivity over a wide linear dynamic range. Although the sensor based on CuW NSs/GCE [45] had a wider working range and demonstrated a lower LOD (1.2 nM), the proposed voltammetric sensors in this work outclass it in terms of sensitivity (1.20 μA μM^–1^ cm^–2^) and peak potential value (+0.33 V). Additionally, the analysis time of our voltammetric sensors is ca. 20 times lower than that of the previously reported sensor (2100 s).

The repeatability of the analytical methods employing the 5-ASA sensors was examined by determining the relative standard deviations of five responses, obtaining 2.5% for PMB_DES_/CNT/GCE and 1.6% for CNT/PMB_DES_/GCE using the CA method and 2.2% for PMB_DES_/CNT/GCE and 4.9% for CNT/PMB_DES_/GCE by means of DPV. The stability of both hybrid sensors was evaluated in two steps. First, 30 consecutive measurements of 50.0 μM 5-ASA were performed, both platforms showed excellent operational and conservation stability, maintaining their electrochemical response (only a 6% decrease). Second, multiple scans in BR buffer were performed with CNT/PMB_DES_/GCE, the sensitivity decreasing only to 93%. All modified electrodes were stored at 4 °C.

### 3.3. Simultaneous Analyses of 5-ASA and Acetaminophen

DPV was selected for the quantification of 5-ASA using the sensor with the best analytical performance (CNT/PMB_DES_/GCE) in the presence of its structural analogue APAP, and for real sample analysis due to its advantages such as ca. seven-fold improved sensitivity as compared with the CA technique, the speed of measurement (100 s), lower detection limit (7.7 nM), low background current, and easy signal processing.

For interference studies, the response of CNT/PMB_DES_/GCE sensor to 5-ASA was tested in the presence of different concentrations of APAP. 

CNT/PMB_DES_/GCE revealed a very good separation of peak potentials (+0.18 V) to achieve the accurate simultaneous determination of 5-ASA and APAP in their mixture. Figure 6a shows the DPVs obtained with CNT/PMB_DES_/GCE for different concentrations (10–70 µM) of 5-ASA in the presence of 10 µM APAP, while Figure 6b shows the corresponding oxidation peaks of APAP (5–60 µM) in the presence of a constant concentration of 5-ASA (10 µM). The hybrid nanostructured sensor revealed a peak-to-peak separation of the analytes; hence, the faradaic currents of 5-ASA were dependently increased in accordance with the bulk concentration in the presence of a constant oxidation peak of APAP corresponding to 10 µM, the sensitivity (9.27 μA cm^−2^ μM^−1^) being very close to that obtained in standard conditions (5.7% decrease).

Furthermore, by keeping the 5-ASA concentration constant, a calibration plot for APAP with very good sensitivity of 12.5 μA cm^−2^ μM^−1^ and low detection limit (2.9 µM) was obtained. The potential value (+0.35 V) corresponding to the APAP oxidation peak value is even lower in the BR buffer (pH 7.0) than that in 0.1 M KCl reported in one of our previous studies [31]. This shows that the hybrid PMB_DES_–CNT film is very sensitive towards simultaneous detection of both analytes.

### 3.4. Real Sample Analysis

The sensor architecture CNT/PMB_DES_/GCE was used under optimized conditions for the analysis of pharmaceutical formulations (Pentasa^®^, 500 mg 5-ASA/tablet). After sample preparation and adequate dilution steps, as previously described in the Materials and Methods Section, the sensor was employed for both CA (Figure 7a,b) and DPV (Figure 7c,d) measurements, obtaining good recovery values. A slightly higher recovery was found with the CA technique (105.2 ± 3.1% RSD), while by DPV analysis was 97.8 ± 3.0% RSD, both being very close to 100%, Table 2. The sensitivity of both methods was ca. 85% of that in standard solutions, showing that the proposed electrochemical sensor provided a good stable response when used in a complex matrix.

In order to further test the application to different pharmaceutical formulations, the sensor was employed for APAP determination. In Coldrex Max Grip^®^ (1 g acetaminophen, 10 mg phenylephrine hydrochloride, 40 mg ascorbic acid), recovery values were 101.2% ± 2.7% for APAP detection in the presence of 50 µM 5-ASA constant concentration. For 5-ASA detection from Pentasa^®^ tablets in the presence of 50 µM APAP, the recovery values were 102.0% ± 3.3%. The apparent recoveries indicate that the proposed electrochemical sensor is efficient for one component determination, and also for simultaneous analysis of 5-ASA and APAP, having an excellent level of reliability for practical applications.

## 4. Conclusions

In this work, the developed CNT/PMB_DES_/GCE electrochemical sensor provides a new, sensitive and selective method for 5-aminosalicylic acid (5-ASA) detection, by utilizing the unique properties of phenazine poly(methylene blue) film synthesized electrochemically in deep eutectic solvents, such as high specific surface area and electrocatalytic properties. The sensing properties were greatly improved by combining the polymer film with carbon nanotubes. The pH of the BR supporting electrolyte and the applied potential in the CA method were optimized and several sensing architectures based on PMB_DES_ and CNT were tested for 5-ASA detection. The CNT/PMB_DES_/GCE sensor was applied to the simultaneous detection of 5-ASA and acetaminophen; the sensitivity towards 5-ASA decreased by only 5.7% in the presence of APAP. The possible application and robustness of this method were confirmed by the determination of 5-ASA in a drug tablet, without any prior complex sample pretreatment or preconcentration at the electrode surface, with CNT/PMB_DES_/GCE, in which recovery values between 97.8% and 105.2% were obtained, demonstrating that this method is promising for application to biological samples.

## Data Availability

Not applicable.

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
