# Peer review of "Hybrid Nanocomposite Platform, Based on Carbon Nanotubes and Poly(Methylene Blue) Redox Polymer Synthesized in Ethaline Deep Eutectic Solvent for Electrochemical Determination of 5-Aminosalicylic Acid"

_sensors, 2021, doi:10.3390/s21041161_

Round 1

Reviewer 1 Report

I think the conjugation of nanotubes with redox polymers synthesised through deep eutectic solvents is innovative in the electroanalysis field. However I do not consider this study totally innovative as the same “formula” was used already for analysis of a similar drug (acetaminophen) and ascorbic acid in a previous study from the same authors (Oana Hosu et al, Nanocomposites based on carbon nanotubes and redox-active polymers synthesized in a deep eutectic solvent as a new electrochemical sensing platform, Microchim Acta (2017) 184:3919–3927).

In general, the manuscript is well written with clear ideas. The overall concept and objectives of the work were clearly understood. The sensors are carefully characterized with all important figures of merit properly presented. Comparisons between different results obtained as well as results from other authors are also given. Figure captions are well informative.

I recommend the manuscript for publication with some minor changes though in my opinion the manuscript is a little bit too long. For instance, some information presented is somewhat repeated in the results and discussion section and probably would suit better as supplementary material in order to simplify the text.

The comments were made in the pdf of the manuscript.

Author Response

We would like to thank the reviewers for the valuable comments and time, we have responded below to their comments.

Reviewer 1

I think the conjugation of nanotubes with redox polymers synthesised through deep eutectic solvents is innovative in the electroanalysis field. However, I do not consider this study totally innovative as the same “formula” was used already for analysis of a similar drug (acetaminophen) and ascorbic acid in a previous study from the same authors (Oana Hosu et al, Nanocomposites based on carbon nanotubes and redox-active polymers synthesized in a deep eutectic solvent as a new electrochemical sensing platform, Microchim Acta (2017) 184:3919–3927).

In general, the manuscript is well written with clear ideas. The overall concept and objectives of the work were clearly understood. The sensors are carefully characterized with all important figures of merit properly presented. Comparisons between different results obtained as well as results from other authors are also given. Figure captions are well informative.

I recommend the manuscript for publication with some minor changes though in my opinion the manuscript is a little bit too long. For instance, some information presented is somewhat repeated in the results and discussion section and probably would suit better as supplementary material in order to simplify the text.

 The comments were made in the pdf of the manuscript.

We thank the reviewer for the comments, we have made some changes throughout the entire manuscript for better clarity according to the reviewers’ comments and suggestions. All changes were highlighted in yellow.

  1. Lines 84-86: This information isn’t too important and so it can be deleted. Probably here is missing the explanation concerning why is important the detection of 5-ASA drug. Is it electrochemically under-characterized? is it important for clinical analysis, industrial analysis, or environmental analysis?

This paragraph was rephrased for better clarity. The electrochemical sensors for 5-ASA were already reported in Table 1. All changes are highlighted in yellow.

  1. The last paragraph of the introduction is usually dedicated to the objectives and a resume of the work performed. Results and conclusions should be avoided in this part: in which RSD values between 97.8% and 105.2% were obtained. The proposed method is simple, sensitive, easy to apply and economical for routine analysis. This phrase probably would suit better on the abstract and not in the introduction.

The comments are appreciated; these paragraphs were revised accordingly.

  1. lines 192-198: I cannot see the relevance of this discussion for the purpose of 5-ASA analysis as well as respective Figures 2e and 2f. Please explain.

As PMB is a redox polymer, the PMB/GCE sensor was electrochemically characterized. The results showed that the PMB does not influence the electrochemical process of 5-ASA as the peak potentials of PMB shift negatively by the same amount as for 5-ASA on increasing the pH. The paragraph was rephrased for better clarity and moved to Supplementary information (Figures S1a and S1b), as a result to one of the below comments.

  1. As clearly said here and seen in Figure 2b, the sensor CNT/PMBDES/GCE achieved the highest peak. Why not use it solely for further optimization studies instead the PMBDES/GCE?

The reviewer made a good point with this comment. First, the PMB/GCE sensor was taken into consideration for determination of the oxidation mechanism of 5-ASA. It was also of interest to compare if there is any influence of the 5-ASA electrooxidation process at bare GCE and PMB/GCE, respectively, as the polymer itself is electroactive. At a later point, the incorporation of CNT was taken into consideration to increase the sensitivity of the sensor.

  1. In order to uniformize the sensor designation, in Figure 2b and Figure 4b change to CNT/PMBDES/GCE and PMBDES/CNT/GCE.

Many thanks for the comment. The figures were uniformized.

  1. In my opinion, the same information is repeated in different figures and the text. Calibration curves in the same conditions for sensor CNT/PMBDES/GCE is present on Figure 3d, Figure 4a and 4b. In order to simplify the text wouldn’t be better to put some results as supplementary material, or simply eliminate?

We thank the reviewer for the comments, we made some changes and reorganization to aid the discussion throughout the results presented. Some figures alongside their discussion were sent to Supplementary information material. Figures 2e and 2f were converted into Figures S1a and S1b and Figures 3c and 3d into Figures S2 a and S2b, respectively. Table 1 was also moved to Supplementary information material.

  1. Horizontal axis on inset of Figures 5a and 5b are wrong. Also, a scale on the vertical axis would be better to perceive the curve equation and thus the sensitivity.

Thank you for the comment; we apologize for the unexpected error. Insets were corrected.

Reviewer 2 Report

This manuscript reports that a novel hybrid composite of conductive poly(methylene blue) (PMB) and carbon nanotubes (CNT) was prepared for the detection of 5-aminosalicylic acid (5-ASA). The main novelty is using CNT/PMBDES/GCE which is potentially very interesting.  However, minor revisions are required before the work is suitable for publication. These include

Comment 1:  P 1, L 28 (from top): “…in the range from 5 μM to 100 μM 5-ASA by DPV method, with a LOD of 7.7 nM,”  Please give the LOD of 7.7 nM Calculation method.

Comment2:  It can be observed that PMB oxidation and reduction peaks both decrease upon 5-ASA addition in Figure 1. Why is this phenomenon observed, what is the principle. Figure 1 should be thinked again very carefully.
Comment 3:   Scientific attitude seriously and the manuscript drafted carefully are required. P 3, L 96 (from top): “….which RSD values between 97.8% and 105.2%”, RSD in this sentence should be revised to "recovery " ;  P 10, L 299 (from top): “Table 2Error! Reference source not found.”, This sentence has to be dealt with.

Author Response

Reviewer 2

This manuscript reports that a novel hybrid composite of conductive poly(methylene blue) (PMB) and carbon nanotubes (CNT) was prepared for the detection of 5-aminosalicylic acid (5-ASA). The main novelty is using CNT/PMBDES/GCE which is potentially very interesting. However, minor revisions are required before the work is suitable for publication. These include

We thank the reviewer for the comments, we have made some changes throughout the entire manuscript for better clarity according to the reviewers’ comments and suggestions. All changes were highlighted in yellow.

Comment 1:  P 1, L 28 (from top): “…in the range from 5 μM to 100 μM 5-ASA by DPV method, with a LOD of 7.7 nM,” Please give the LOD of 7.7 nM Calculation method.

The detection limit was calculated as 3 times the standard deviation of the calibration plot divided by the slope of the calibration plot (Brett, C. M. A.; Oliveira-Brett, A.-M. Electroanalysis; Oxford University Press: Oxford) as further reported in the results and discussion section. The calculation method was inserted in the abstract.

Comment2: It can be observed that PMB oxidation and reduction peaks both decrease upon 5-ASA addition in Figure 1. Why is this phenomenon observed, what is the principle? Figure 1 should be thinked again very carefully.

The reviewer made a good point with this comment. Indeed, there is a decrease of the PMB anodic and cathodic peak intensity upon 5-ASA addition that might be due to the high concentration of 5-ASA present in the solution that can impede the redox reaction at the polymer interface. Moreover, the –COOH group in the 5-ASA structure is prone to form hydrogen bonds with the nitrogen atoms in the polymer film. This interaction has two results: i) increases the 5-ASA concentration at the PMB surface, which will improve 5-ASA reaction kinetics, advantageous for sensing and ii) hinders the electrochemical process of PMB, due to a decrease in counter ion density/availability at the polymer/solution interface. Similar interaction between the amino groups of PMB films and hydrocarbyl groups of analytes was reported in “Facile preparation of poly(methylene blue) modified carbon paste electrode for the detection and quantification of catechin”, Materials Science and Engineering: C, 73, 2017, 552-561.

Comment 3: Scientific attitude seriously and the manuscript drafted carefully are required. P 3, L 96 (from top): “….which RSD values between 97.8% and 105.2%”, RSD in this sentence should be revised to "recovery " ;  P 10, L 299 (from top): “Table 2Error! Reference source not found.”, This sentence has to be dealt with.

The comments are appreciated. The phrases were revised accordingly and all changes are highlighted in yellow. Corrections have been made throughout the entire manuscript to improve its scientific quality.
